# Temporal Dynamics of the Goose Habitat in the Middle and Lower Reaches of the Yangtze River

Ke He [1,2,†], Jialin Lei [1,†], Yifei Jia [1], Entao Wu [1], Gongqi Sun [1,3], Cai Lu [1], Qing Zeng [1] and Guangchun Lei [1,*]

1 Center for East Asian-Australasian Flyway Studies, Beijing Forestry University, Beijing 100083, China; heke0611@bjfu.edu.cn (K.H.); leijialinbjfu@bjfu.edu.cn (J.L.); jiayifei@bjfu.edu.cn (Y.J.); entaowu@bjfu.edu.cn (E.W.); sungongqi@foxmail.com (G.S.); lucai.wetland@foxmail.com (C.L.); zengqing@bjfu.edu.cn (Q.Z.)

2 Ministry of Education Key Laboratory of Southwest China Wildlife Resource Conservation, China West Normal University, Nanchong 637002, China

3 Academy of Inventory and Planning National Forestry and Grassland Administration, Beijing 100013, China

* Correspondence: guangchun.lei@foxmail.com; Tel.: +86-010-6233-6717

† These authors contributed equally to this work.

**Abstract:** The middle and lower reaches of the Yangtze River are the most important areas for geese to overwinter in the East Asian–Australasian Flyway, where about 180,000 geese fly to overwinter each year. Over the past 20 years, the region has experienced extensive and rapid land cover changes that may have exceeded the adaptability of geese, and have led to suitable goose habitat area loss, thereby, reducing the stability of the geese population. In order to identify the suitable goose habitat areas in this region, based on ensemble modeling and satellite tracking data, in this study, we simulated the spatial distribution changes in the suitable goose habitat areas over the past 20 years. The results showed that the suitable goose habitat areas had suffered varying degrees of loss, among which, the lesser white-fronted goose had the greatest suitable goose habitat area loss of over 50%. Moreover, we found that wetlands, lakes, and floodplains were the key components of suitable goose habitat areas, and the categories (land use) showed significant differences in different periods ($p < 0.01$). This may be one of the main reasons for the decrease in suitable goose habitat areas. The results of this study provide an important reference for the adaptive management and protection of geese in the middle and lower reaches of the Yangtze River.

**Keywords:** habitat loss; geese; species distribution models (SDMs); land use change; middle and lower reaches of the Yangtze River

## 1. Introduction

Habitat loss and degradation has been a major cause of wildlife population decline [1–5]. In the nonbreeding season (overwintering period), goose habitat areas mainly consist of floodplains [1,6–9]. While floodplains are highly complex and dynamic ecosystems, they are also among the most threatened ecosystems because they are often dominated by humans and may experience a high intensity of anthropogenic activity [10–14].

The middle and lower reaches of the Yangtze River (MLYR) is one of the most important freshwater ecoregions in the world [15,16]. The numerous lakes connected to the Yangtze River (such as Poyang Lake and Dongting Lake in this region) form a complex river-lake relationship with the Yangtze River, creating extremely rich wetland ecosystem types (nine Ramsar sites of international importance, http://www.ramsar.org/pdf/sitelist.pdf, accessed on 11 October 2021) and providing a habitat for many important and endangered waterbirds [17–19]. The population of geese accounts for about 35% of the total number of waterbirds in the MLYR. This area constitutes the most important overwintering site for geese in the East Asian–Australasian Flyway (EAAF), with nearly 180,000 geese overwintering there every year according to a 2004 survey of birds by [20]. The geese that overwinter

in this area include the lesser white-fronted goose (LWFG, Anser erythropus), the greater white-fronted goose (GWFG, Anser albifrons), the bean goose (BG, Anser fabalis), the swan goose (SG, Anser cygnoides), and the greylag goose (GG, Anser anser), among which the LWFG and SG have been recognized as vulnerable by the International Union for Conservation of Nature (IUCN). The LWFG, SG, and GG in the EAAF generally overwinter in this region [21–23], while 20% of GWFGs and 70% of BGs overwinter in this area, and the other geese overwinter in Japan and Korea [24,25]. The population of these five species of geese accounts for more than 99% of the total number of geese in the MLYR (unpublished data from the Center for East Asian–Australasian Flyway Studies).

Waterbirds represent an important environmental indicator group, especially for the status of wetland ecosystems in the MLYR [26]. Over the past 20 years, the MLYR has become one of the regions with the fastest economic growth in China. Human activities in the region have strongly disturbed the hydrological rhythm, especially the unreasonable development and utilization of lakes, wetlands, and floodplains, as well as the cascade development of hydropower stations in the Yangtze River Basin, such as the Three Gorges Dam. As a result, the loss and degradation of goose habitat areas have resulted in a sharp decline in the population of geese in the region [7].

Over the past two decades, species distribution models (SDMs) have been widely used to study species spatial distribution patterns and guide conservation planning [27,28]. SDMs can be adapted to different spatial resolutions, and the available data sources can help researchers to understand the population distribution of species and can provide valuable insights even for species that are rarely studied [29]. Currently, common SDMs include the generalized linear model (GLM) [30], random forest (RF) [31], and maximum entropy (MAXENT) [21]. Each SDM has different characteristics and advantages. Therefore, an increasing number of studies have used ensemble modeling to integrate the advantages of various SDMs to study the spatial distribution of species [27,32]. By combining models with different assumptions and algorithms, the integrated model can provide more robust results than a single model [33].

In this study, SDMs combined with GPS satellite tracking data were used, for the first time, to study the large-scale biogeography of five species of geese in the MLYR, aiming to determine the main environmental variables affecting their habitat areas, and to evaluate their habitat conditions in different periods and the change trends of their habitat areas, which is of great scientific significance to the protection of geese in the research area.

## 2. Data and Methods

### 2.1. Study Area

The Yangtze River, the longest river in Asia and the third longest river in the world [34], is unique in its extensive transitory basin wetlands. The wetlands are replenished by summer monsoon rains, bringing nutrient-rich and sediment-rich water, followed by falling water levels in autumn and winter [34]. The MLYR, from the Three Gorges Dam to the estuary, mainly covers most of Hubei, Hunan, Jiangxi, Anhui, Jiangsu, Zhejiang, and Shanghai, as well as some regions of Guangxi and Henan, with a watershed area of about 800,000 km$^2$ (Figure 1) [35].

### 2.2. Data and Model

For comparative analysis, we divided the 20 years from 2000 to 2019 into four periods, namely, Period 1 from 2000 to 2004, Period 2 from 2005 to 2009, Period 3 from 2010 to 2014, and Period 4 from 2015 to 2019.

The calculation results of geese in Period 4 were used as the current distribution, and then, the final ensemble model was projected to the past by using the occurrence and environment data of other periods.

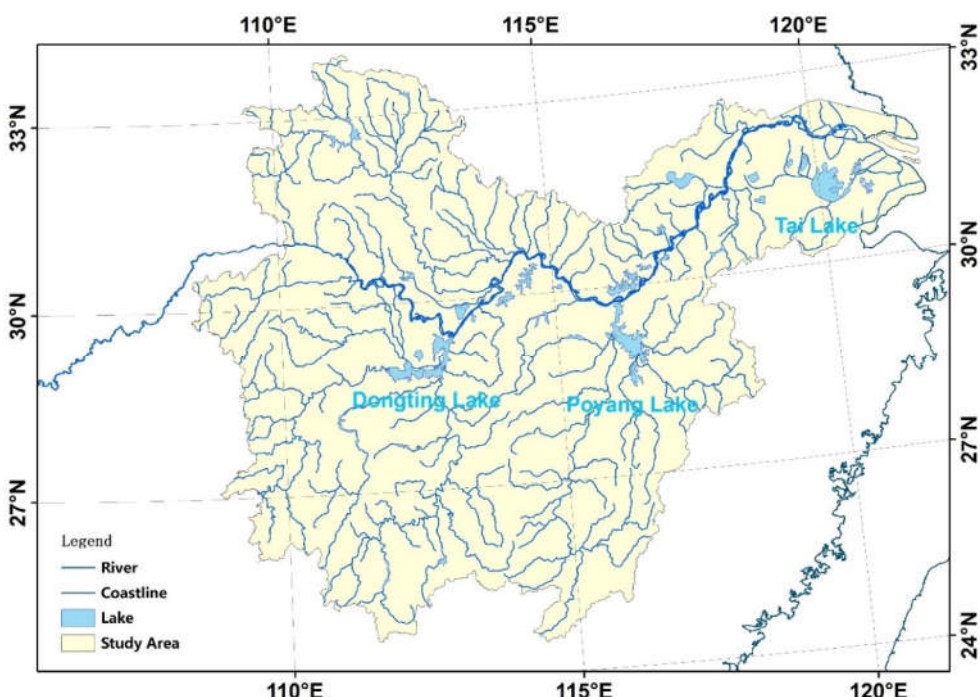

**Figure 1.** Geographical location of the study area.

### 2.2.1. Goose Occurrence Data

Since 2015, GPS trackers have been applied to 141 geese, and by 2019, 403,811 goose occurrence data were obtained. Because the MLYR is the overwintering grounds for geese, we only selected the occurrence data from October to December and from January to March. In order to avoid interference with the model due to differences in sites during the migration of geese, we eliminated all sites with velocities greater than 1 [1]. In order to reduce the error of occurrence data in geographical coordinates and reduce the sampling bias effect of the occurrence dataset, the occurrence data were compiled at a spatial resolution of $1 \times 1$ km [27]. After removing duplicate records within each grid cell, we obtained 2664 presence records to simulate the habitat areas of these five goose species (Table 1).

**Table 1.** Number of occurrence data points for five species of geese (lesser white-fronted goose (LWFG), greater white-fronted goose (GWFG), bean goose (BG), swan goose (SG), and greylag goose (GG)).

| Species | Number of Occurrence Data Points |
|---------|:--------------------------------:|
| LWFG | 419 |
| GWFG | 752 |
| BG | 852 |
| SG | 487 |
| GG | 154 |
| Total | 2692 |

### 2.2.2. Climate Data

The data of climate variables are important for predicting species distribution, especially for analyses over long time spans [36,37]. The climate variables used in this study were derived from CHELSA (http://chelsa-climate.org, accessed on 5 September 2021) [38,39], mainly using the three variables of monthly rainfall (PRE), monthly average maximum temperature (TMAX), and monthly average minimum temperature (TMIN). The time period was 2000–2018, although the precipitation data for 2018 were missing, and the climate data accuracy was 30 arc seconds (about 1 km$^2$). In order to effectively assess the goose habitat in the MLYR, only the data during the overwintering period were selected in

this study. For example, the averages of October to December of the first year and January to March of the next year were taken as one data point (the rainfall in 2000 was the average of the rainfall from October to December of 1999 and January to March of 2000).

### 2.2.3. Normalized Difference Vegetation Index (NDVI) and Normalized Difference Water Index (NDWI)

The NDVI and NDWI have been widely used to evaluate the distribution of geese [7,8,40]. Our NDVI and NDWI data were derived from the "Landsat 7 Collection 1 Tier 1 1 8-Day NDVI/NDWI Composite" database in the Earth Engine Data Catalog. At the same time, we used the Google Earth Engine platform (https://code.earthengine.google.com/, accessed on 15 August 2021) to download directly for the period 2000–2019. These data have a resolution of 30 m. In order to ensure the consistency of the data time, we adopted the same processing method as that used for climate data.

### 2.2.4. Land Use

The use of land use data plays an important role in predicting species distribution at large landscape scales [41–43]. The land use data used in this study were obtained from the Data Center for Resources and Environment of the Chinese Academy of Sciences (http://www.resdc.cn/, accessed on 20 August 2021) and included cultivated land, forest land, grassland, water areas, residential land, and unused land as six primary types, with 20 secondary types. These data have a resolution of 1 km$^2$. Based on the needs of this study, we selected four primary types and 12 secondary types (Table 2). In this study, the land use data of the first year of each period were selected as the land use data of the period, that is, the land use date of Period 1 was 2000 (it contains 4 years of land use data, 2000, 2005, 2010, and 2015, respectively, Appendix A, Figure A1). Because the land use data were the classification variable and the partial SDM model was not conducive to the classification variable, in order to conduct the quantitative analysis, in this study, we transformed 12 types of land use into a continuous variable using Euclidean distance. For example, wetlands were transformed into the distance to wetland (dis_wl).

**Table 2.** Land use classification.

| Level 1 | Level 2 | Meaning |
|---|---|---|
| Cropland | Paddy Field (dis_pf) | Cropland with a guaranteed water source and irrigation facilities that can be irrigated normally in normal years and used to grow rice, lotus root, and other aquatic crops. |
| | Upland Field (dis_uf) | Cropland without an irrigation water source or facilities that depends on natural water to grow crops; dry-crop-cultivated land with a water source and irrigation facilities that can be irrigated normally in normal years; cultivated land mainly used for vegetable cultivation. |
| Water | River (dis_ri) | Land below the perennial water level of rivers and main rivers formed by natural or artificial excavation. Artificial channels include an embankment. |
| | Lake (dis_la) | Land below the perennial water level in a natural water accumulation area. |
| | Reservoir (dis_re) | Land below the perennial water level in an artificial water storage area. |
| | Mudflat (dis_mf) | The tidal zone between the high tide level and the low tide level of the coastal spring tide. |
| | Floodplain (dis_fp) | Land between the water levels of rivers and lakes in normal seasons and those in flood seasons. |

**Table 2.** *Cont.*

| Level 1 | Level 2 | Meaning |
|---|---|---|
| Construction land | Urban Land (dis_ul) | Land used in large, medium, and small cities and built-up areas above the county level. |
| | Rural Land (dis_rl) | Rural settlements that are independent of cities and towns. |
| | Other Construction Land (dis_ocl) | Land used for factories and mines, large industrial areas, oil fields, salt fields, quarries, traffic roads, airports, and other construction land uses. |
| Unused Land | Wetland (dis_wl) | Land with flat and low-lying terrain, poor drainage, long-term moisture, seasonal water accumulation or perennial water accumulation, and surface growth of hygrophytes. |
| | Bare Land (dis_bl) | Land covered by surface soil where the vegetation coverage is less than 5%. |

### 2.2.5. Elevation Data

The elevation data were obtained from a digital elevation model (DEM) with a 30 m resolution and downloaded from the International Scientific and Technical Data Mirror Site, Computer Network Information Center, Chinese Academy of Sciences (http://www.gscloud.cn, accessed on 15 August 2021) [44]. To match the resolution of climate variables, the DEM data were resampled at 1 km$^2$ resolution using a bilinear interpolation

### 2.2.6. Model

We used the stacked species distribution model (SSDM) software package in R software to simulate the suitable goose habitats [45,46]. For this purpose, we used seven species distribution models for the calculations: the GLM, RF, support vector machines (SVM) [47], artificial neural network (ANN) [48], generalized additive model (GAM) [49], classification tree analysis (CTA) [50], and generalized boosting model (GBM) [51].

To evaluate the accuracy of each algorithm, we performed 10 cross-validations for each algorithm; 70% of each dataset was used as training data and the remainder was used to test algorithm performance. The area under the receiver operating characteristic curve (AUC) [27,29,46] was used to evaluate the goodness-of-fit of each model. When the AUC value of the model was greater than 0.9, it was considered to be an excellent fit; when the AUC was 0.9–0.8, it was considered to be a good fit; when the AUC was 0.8–0.7, it was regarded as an acceptable fit; and when the AUC was less than 0.7, the model was regarded as a poor fit [28]. The habitat suitability maps were converted to binary presence absence maps using a threshold that maximums model sensitivity plus specificity [27].

To avoid possible multicollinearity leading to biased model estimates, we tested Pearson correlations between environmental factors and defined the absolute value of the correlation coefficient R > 0.7 as a threshold [27]. Finally, we selected 18 variables, such as land use and climate, among which the correlation between TMAX and TMIN was 0.9. Because these were important indicators for predicting suitable goose habitat, they were also included in the analysis (Appendix A, Figure A2).

All environmental variables for this study were processed using ArcGis 10.6 in order to obtain a uniform resolution and coordinate system. The comparative analysis of all results was completed in R (version 4.1.1) software.

## 3. Results

### 3.1. Model Performance and Variable Contribution

It was found that the seven algorithms used for species distribution models had excellent recognition abilities for the five species of geese, and the average AUC values of LWFG, GWEG, SG, and BG were higher than 0.9. The results showed that the models had excellent fits, with AUC values of 0.944 ± 0.002, 0.938 ± 0.002, 0.930 ± 0.002, and

0.920 ± 0.002 for the LWFG, GWEG, SG, and BG models, respectively. The average AUC value of GG was 0.886 ± 0.004, indicating that the model fit was good (Figure 2). Overall, among the seven model algorithms, SVM had the lowest AUC value (0.916 ± 0.031), while GAM had the highest AUC value (0.934 ± 0.028).

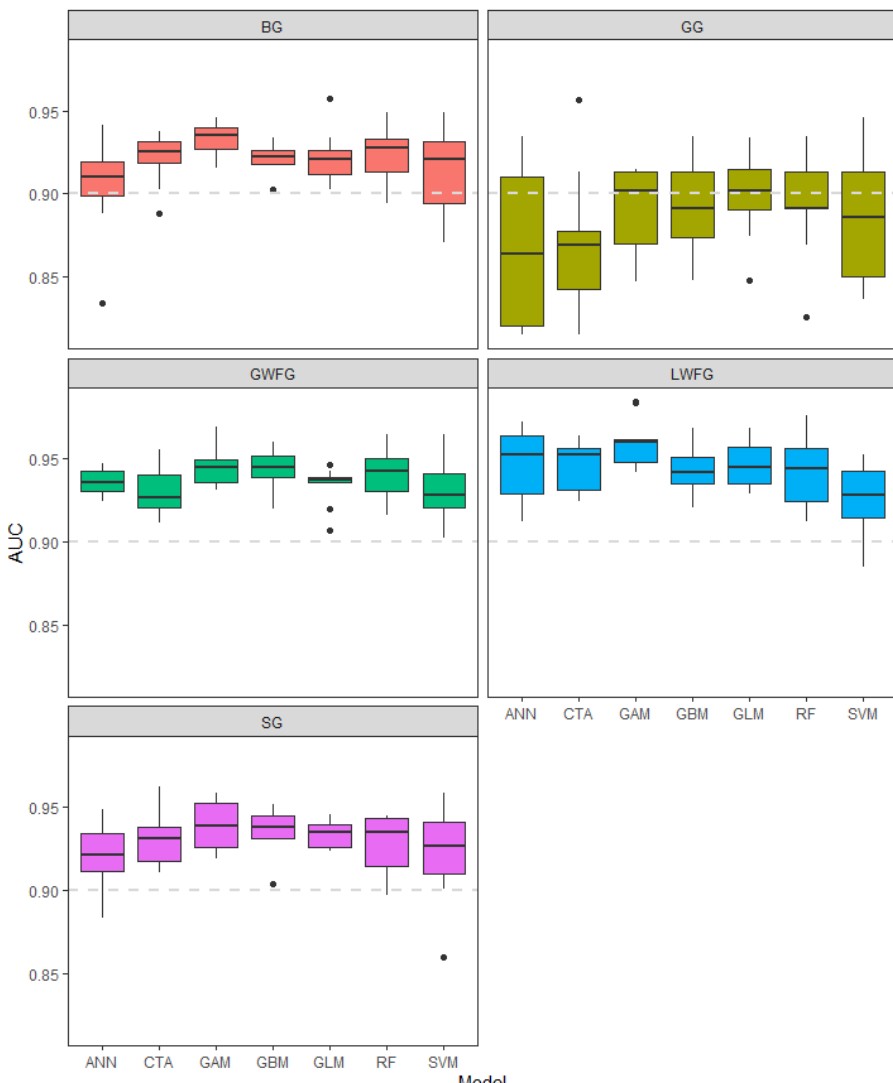

**Figure 2.** Box plots of interquartile range (IQR), range, and median model performance of the seven modeling algorithms used to predict the habitat suitability of five goose species. The dots are potential outliers which are greater than the 75th percentile + 1.5 IQR or less than the 25th percentile − 1.5 IQR. The medians are represented by thick black lines. Generalized linear model (GLM); random forest (RF); support vector machines (SVM); artificial neural network (ANN); generalized additive model (GAM); classification tree analysis (CTA); generalized boosting model (GBM); lesser white-fronted goose (LWFG); greater white-fronted goose (GWFG); bean goose (BG); swan goose (SG); and greylag goose (GG).

The results showed that land use data contributed the most to the simulation of suitable goose habitat areas, with an average contribution rate of 0.781 ± 0.009, followed by climate data, with a contribution rate of 0.097 ± 0.009. Altitude, NDVI, and NDWI contributed less at 0.053 ± 0.008, 0.045 ± 0.011, and 0.023 ± 0.004, respectively (Appendix B, Table A1). Specifically for each variable, among all 18 variables, the contributions of dis_la, dis_fp, dis_wl, altitude, NDVI, and TMIN were more than 0.05, and the contributions of dis_la, dis_fp, and dis_wl to all the goose habitat areas were more than 0.05 (Figure 3). Although all geese have a high demand for dis_la, dis_fp, and dis_wl, the degree of their specific

needs varies. The contributions of dis_wl to the suitable habitat of LWFG, GWFG, and SG geese were $0.294 \pm 0.022$, $0.238 \pm 0.019$, and $0.225 \pm 0.017$, respectively. The contributions of dis_la to the suitable habitat of BG and GG were the highest, at $0.219 \pm 0.015$ and $0.268 \pm 0.018$, respectively (Appendix B, Table A1).

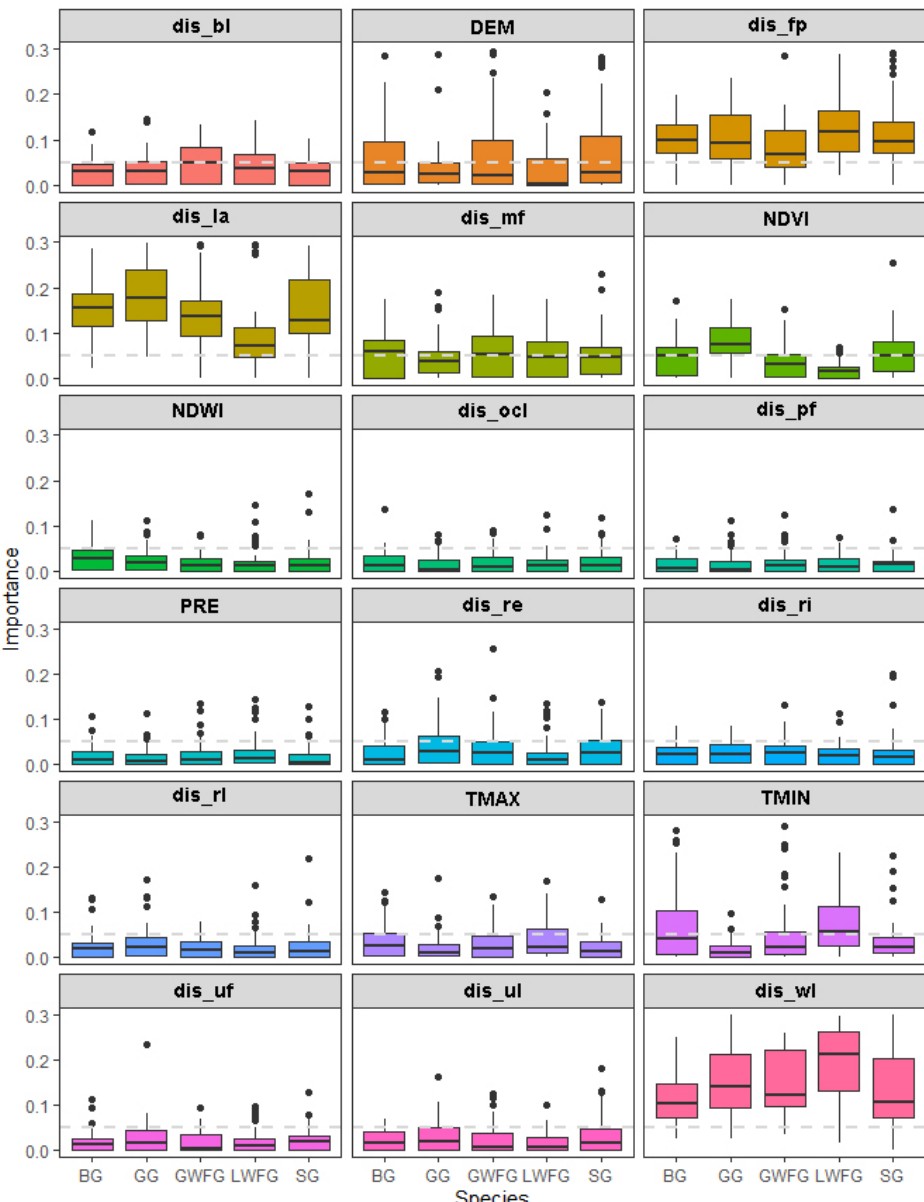

**Figure 3.** Box plots of interquartile range (IQR), range, and median model performance of algorithms used to predict the variable contribution to the model. The dots are potential outliers which are greater than the 75th percentile + 1.5 IQR or less than the 25th percentile − 1.5 IQR. The medians are represented by thick black lines. Lesser white-fronted goose (LWFG); greater white-fronted goose (GWFG); bean goose (BG); swan goose (SG); and greylag goose (GG).

Because dis_la, dis_fp, and dis_wl are important for predicting suitable goose habitat areas, we compared the differences in these three variables in different periods. This study found that dis_fp and dis_la showed a downward trend from Period 1 to Period 4, while dis_wl showed an upward trend (Appendix B, Table A2). The results showed that the three variables exhibited significant differences in the four periods (ANOVA test, $p < 0.01$, Figure 4).

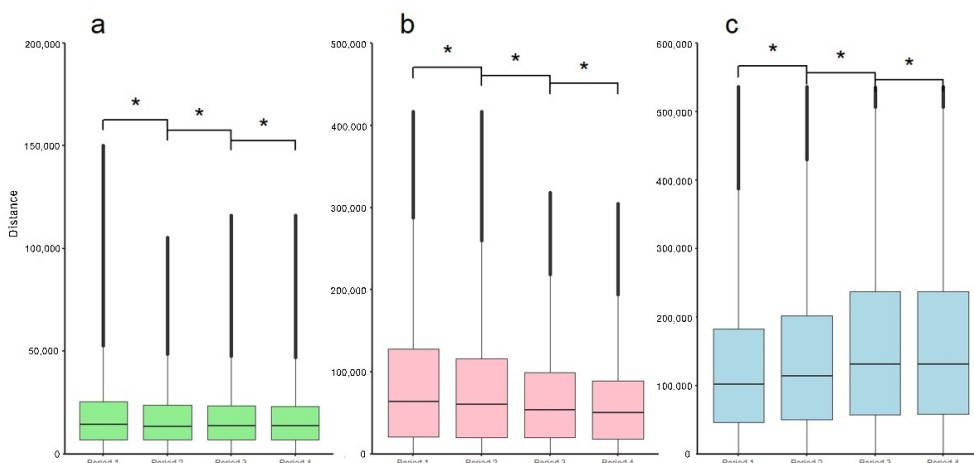

**Figure 4.** The differences of the three variables in the four periods: (**a**) dis_fp; (**b**) dis_la; (**c**) dis_wl. Period 1 is 2000–2004, Period 2 is 2005–2009, Period 3 is 2010–2014, and Period 4 is 2015–2019. The asterisks indicate a significant difference.

### 3.2. Suitable Habitat

Using the habitat classification thresholds (LWFG 0.48, GWFG 0.41, BG 0.44, SG 0.43, and GG 0.39) calculated by the SSMD model, we classified the suitable and unsuitable goose habitat areas (Figure 5). The results showed that the largest suitable goose habitat area for all geese was found during Period 1, and the suitable goose habitat areas for all geese were mainly distributed in the Dongting Lake and Poyang Lake areas, as well as the areas near the mainstream of the Yangtze River between the two lakes. In the same period, the suitable goose habitat area of GG was the largest among the five species of geese, while the suitable goose habitat area of LWFG was the smallest. In terms of suitable goose habitat area loss, the suitable goose habitat area of all geese has declined over the past 20 years, with the LWFG losing the most suitable goose habitat area and GG losing the least suitable goose habitat area. Specifically, LWFG and GFWG decreased the most from Period 3 to Period 4, with a loss of 15,905 km$^2$ (45.85%) and 9217 km$^2$ (23.26%) of suitable habitat area, respectively; BG and GG lost 7191 km$^2$ (14.97%) and 2550 km$^2$ (3.88%) of suitable habitat area from Period 2 to Period 3, respectively. SG lost the most suitable habitat area from Period 1 to Period 2 (3926 km$^2$, 9.75%, Table 3).

**Table 3.** The suitable habitat area (km$^2$) loss and the relative change ratio (%) of suitable habitat between two consecutive periods. Lesser white-fronted goose (LWFG); greater white-fronted goose (GWFG); bean goose (BG); swan goose (SG); and greylag goose (GG).

| Species | Suitable Habitat Area/Change | Period 1 | Period 1 vs. Period 2 | Period 2 vs. Period 3 | Period 3 vs. Period 4 |
|---|---|---|---|---|---|
| LFWG | Suitable habitat | 37,872 | | | |
| | Lost suitable habitat | | −2922 | −260 | −15905 |
| | Relative change ratio | | −7.72% | −0.74% | −45.85% |
| GFWG | Suitable habitat | 46,067 | | | |
| | Lost suitable habitat | | −2747 | −3699 | −9217 |
| | Relative change ratio | | −5.96% | −8.54% | −23.26% |
| BG | Suitable habitat | 52,613 | | | |
| | Lost suitable habitat | | −4576 | −7191 | −4441 |
| | Relative change ratio | | −8.70% | −14.97% | −10.87% |
| SG | Suitable habitat | 40,253 | | | |
| | Lost suitable habitat | | −3926 | −1591 | −2757 |
| | Relative change ratio | | −9.75% | −4.38% | −7.94% |
| GG | Suitable habitat | 67,697 | | | |
| | Lost suitable habitat | | −1972 | −2550 | −1554 |
| | Relative change ratio | | −2.91% | −3.88% | −2.46% |

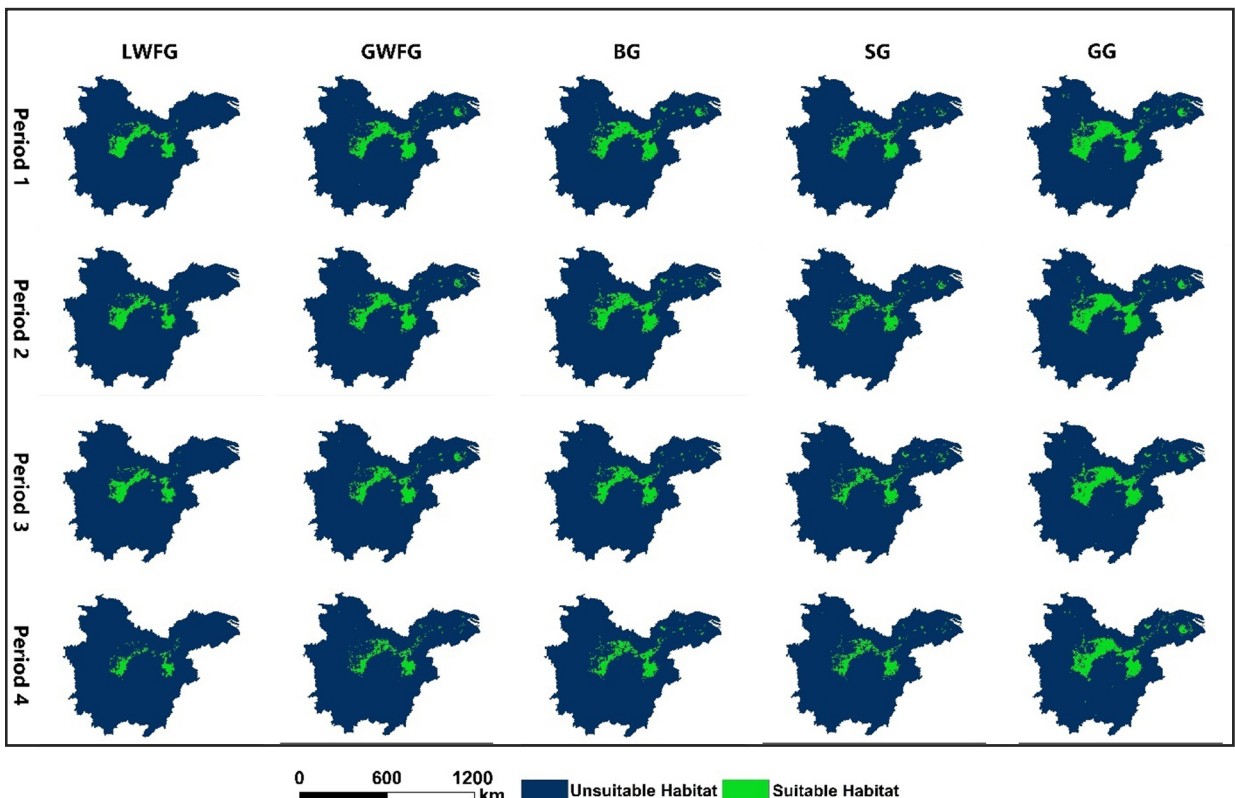

**Figure 5.** Suitable and unsuitable goose habitat areas in the MLYR for the four periods. Period 1 is 2000–2004, Period 2 is 2005–2009, Period 3 is 2010–2014, and Period 4 is 2015–2019. Lesser white-fronted goose (LWFG); greater white-fronted goose (GWFG); bean goose (BG); swan goose (SG); and greylag goose (GG).

In order to elucidate the overall situation of suitable goose habitat loss, we compared suitable goose habitat areas in two periods: Period 1 and Period 4. The results showed that during the 20 years from 2000 to 2019, the suitable goose habitat area of LWFG increased by 486 km$^2$ in some areas and decreased by 19,573 km$^2$ in other areas, with a total suitable habitat area loss of 50.40%, which was the greatest suitable goose habitat area loss among the five species; the suitable goose habitat area of GWFG increased by 1163 km$^2$ and decreased by 16,826 km$^2$, with a total suitable habitat area loss of 34.00%; the suitable goose habitat area of BG increased by 817 km$^2$ and decreased by 17,025 km$^2$, with a total suitable habitat area loss of 30.81%; the new suitable goose habitat area of SG was 3076 km$^2$, while it decreased by 8593 km$^2$, with a total suitable goose habitat area loss of 13.71%; the new suitable goose habitat area of GG was 4688 km$^2$, while it decreased by 8792 km$^2$, with a total suitable goose habitat area loss of 6.24%. GG was the species with the least suitable goose habitat area loss among the five species of geese (Figure 6).

From 2000 to 2019, the peripheries of the suitable goose habitat area for all geese were lost to varying degrees, and the greatest suitable goose habitat area loss was in the marginal areas. The LWFG lost most of its habitat area, except for the Dongting Lake and Poyang Lake areas, especially the suitable habitat area in the northwest of Dongting Lake. Most of the suitable GWFG habitat in the Tai Lake area was lost, and the GWFG habitat in the periphery of the two lakes was also lost. Similar to the GWFG, the BG lost suitable habitat in the Tai Lake area. Moreover, most of the suitable habitat in the peripheries of the two lakes were also lost. Much of the suitable SG habitat in the peripheries of the two lakes was lost as well, but it had also increased in some areas, especially in the lower reaches of the Yangtze River. Among all the geese, the GG had the least suitable goose habitat area loss, which was mainly concentrated in the peripheries of the two lakes, while it had a large increase in suitable goose habitat area in the lower reaches of the Yangtze River (Figure 7).

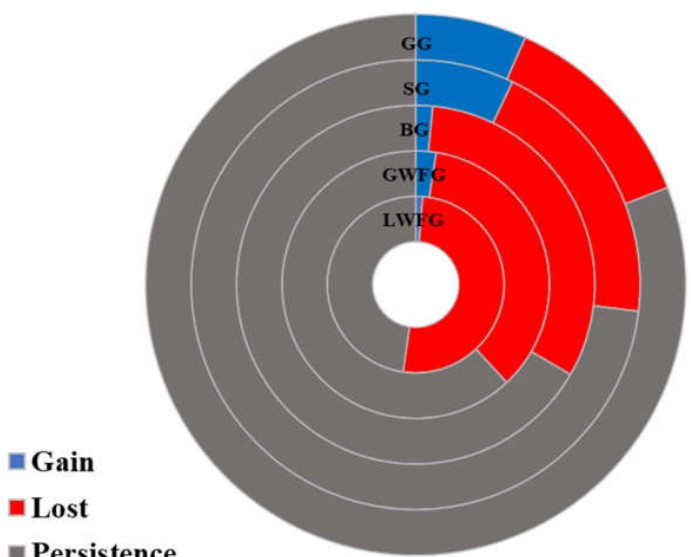

**Figure 6.** The suitable goose habitat loss over the past 20 years. From inside to outside, lesser white-fronted goose (LWFG); greater white-fronted goose (GWFG); bean goose (BG); swan goose (SG); greylag goose (GG).

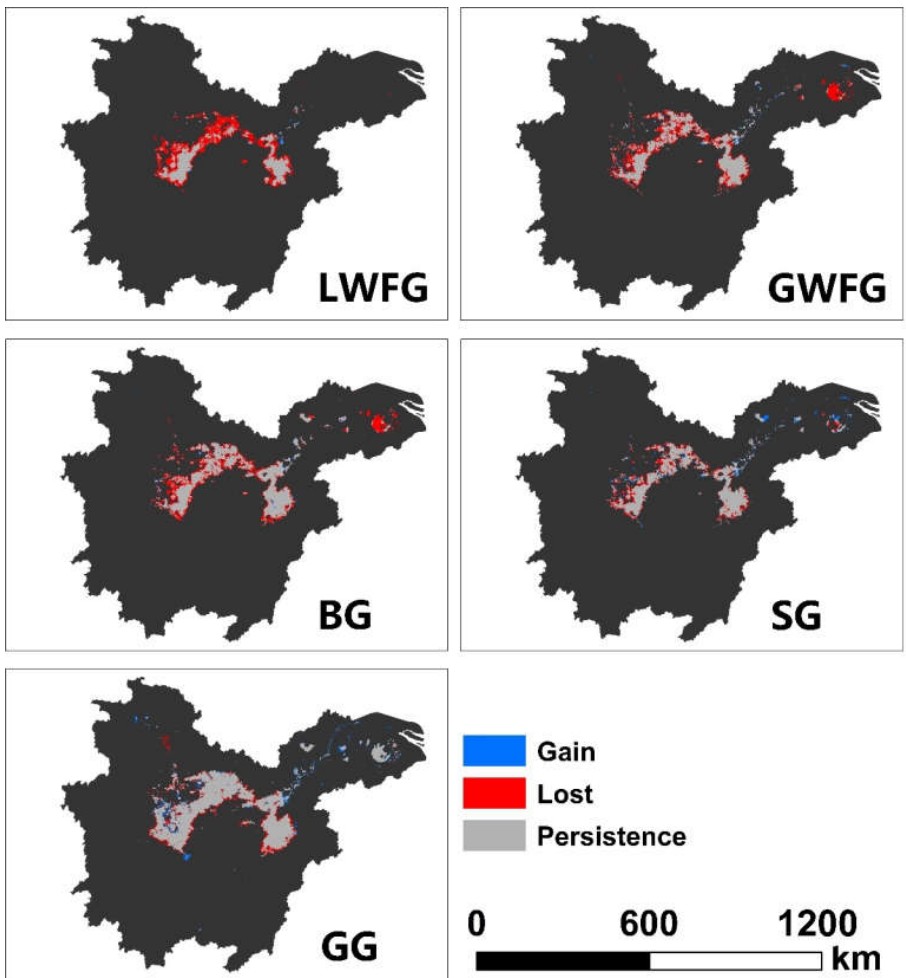

**Figure 7.** Change in suitable goose habitat during Period 1 and Period 4. Lesser white-fronted goose (LWFG); greater white-fronted goose (GWFG); bean goose (BG); swan goose (SG); and greylag goose (GG).

## 4. Discussion

SDMs are widely used to address a variety of ecological problems, including predicting the geographic range of species, assessing the impact of biological invasions, and developing conservation strategies [52–54]. However, some studies have shown that a single-species distribution model has greater uncertainty, while an ensemble model can provide more accurate results [55–57]. In this study, seven SDMs (ANN, GAM, GLM, RF, CAT, GBM, and SVM) were used to combine the individual SDMs with the SSDM model, combined with GPS tracking data and environmental data of five goose species from 2015 to 2019. The habitat adaptability and spatial distribution of these geese in the MLYR over the past 20 years were mapped. These findings were basically in line with the scientific consensus on the habitats of these five species, and a large number of areas lost in previous surveys were also found. These results are of great significance for supporting the protection of goose habitats in the MLYR.

The study area is the most important overwintering site for geese on the EAAF. The goose habitat loss in this area means a decrease in food availability and energy reserves, which lead to a series of problems, such as an increase in intraspecific and interspecific competition pressure, a decrease in the subadult survival rate, and an increase in mortality during migration. Over the past 20 years, the suitable goose habitat areas in the MLYR have been reduced to different degrees. Among the five species studied, LWFG, GWFG, and BG lost more than 16,000 km$^2$ of suitable habitat area (accounting for 50.40%, 34.00%, and 30.81% of their original habitat areas, respectively). However, the suitable habitat areas of SG and GG only decreased by about 8000 km$^2$ (13.71% and 6.24%, respectively). As compared with LWFG, GWFG, and BG, the suitable habitat area losses for SG and GG have had less impact on these species. Zhang et al. [7] found that large geese showed better adaptability to environmental changes, while small geese had a smaller range of feeding habits due to the length and hardness of their beaks, which made small geese more sensitive to environmental changes. This may be one of the reasons why the LWFG had the greatest suitable habitat area loss and GG had the least suitable habitat area loss. The most direct manifestation of habitat loss is a decline in population size, which has declined significantly for LWFG, GWFG, and BG over the past 20 years. In particular, the eastern population of the LWFG (all of which overwinter in the MLYR) decreased from 65,000 geese in the 1980s to 4020 geese in 2020 [58], and its suitable habitat was the most reduced of all five species. The suitable habitat area losses of SG and GG were less, and their populations were relatively stable in the MLYR. This also reflects that habitat area plays an extremely important role in the stability of goose population.

Our results show that the suitable goose habitat areas in the MLYR are shrinking to the area around the two lakes (Figure 7). Many reports in the literature have also reflected this finding [7,22,23,26]. The reason for this phenomenon may be that the two lakes are the two largest freshwater lakes in China and provide abundant food resources for geese. In addition, there are many nature reserves in the region, with large, protected areas and low human disturbance, which makes the habitat of this region better than that of other regions. The vast waters and floodplains in the two lakes areas are also the largest and most complete natural wetlands in the MLYR, and geese overwintering in China prefer natural wetlands [59,60].

A large number of research results have shown that wetlands, floodplains, and lakes were the main components of goose habitat areas in the MLYR [22,23,25,26,59]. In the present study, it was found that dis_wl, dis_fp, and dis_la were very important for predicting the suitable habitat areas of the five species of geese. However, these three variables have changed dramatically over the past two decades, and there were significant differences in the four periods (Figure 4). These changes have posed significant challenges for geese and have led to the loss of suitable habitat areas for geese with poor adaptability. These three variables are closely related to the hydrological rhythm of the Yangtze River [35,61], and the MLYR is a typical case [7]. During the first 20 years of the 20th century, the economy of the MLYR developed rapidly. To satisfy the high demand for electricity, the Yangtze

River was used for hydropower generation. As of 2019, there were 159 hydropower stations with an annual power generation of more than 300 thousand kilowatts in the Yangtze River Basin [62]. This high quantity of hydropower stations has significantly changed the hydrological rhythm of the Yangtze River Basin, leading to changes in wetlands, floodplains, and lakes [35] that have been disastrous for geese. Some studies have concluded that the construction of hydropower stations has destroyed the natural hydrological rhythm, resulting in early or delayed recession of floodplains in the MLYR, thus, affecting the timing and trends of food growth. These phenological changes no longer match the time when geese arrive at overwintering sites, and therefore geese are unable to obtain food in their original habitat areas [7]. The MLYR is the main grain-producing area in China, with well-developed agriculture; the lake areas in the region are shrinking due to reclamation and the demand for agricultural irrigation. The region also exhibits rapid economic development, a large population, rapid expansion of urban areas, rapid growth of tourism and other tertiary industries, and a large number of natural wetlands that have been exploited, which may be one of the reasons for the greater loss of suitable habitat margins for geese [63,64].

This study found that the use of land use data played an important role in simulating suitable goose habitat in the MLYR, with an average contribution rate as high as $0.781 \pm 0.009$ (Appendix B, Table A1). The contributions of climate (PRE, TMAX, and TMIN), DEM, NDVI, and NDWI were relatively small at $0.032 \pm 0.004$, $0.053 \pm 0.008$, and $0.045 \pm 0.011$, and $0.023 \pm 0.004$, respectively (Appendix B, Table A1). This was different from the results of many studies on goose habitat areas, some of which found that hydrological changes were the key factors in changes in goose distribution [65], and some scholars have indicated that food resources were important limiting factors [66,67]. These contrasting results are due to the differences in the scale of the study areas, and the use of land use data plays an extremely important role in predicting the distribution of species at a large landscape scale [41–43]. Studies that have suggested food resources or hydrological changes were the key factors affecting the distribution of geese have mostly been based on small spatial scales, and our research area covers the whole middle and lower reaches of the Yangtze River. Therefore, the difference in spatial scales is an important reason for the differences between our results and those of other studies.

Over the past 20 years, the habitat of wild geese overwintering in the MLYR has experienced different degrees of loss, and suitable goose habitat area has been significantly reduced, resulting in a significant decline in their population. This decline has mainly been caused by human activities. The MLYR is among the areas with the fastest economic development in China, but this rapid economic growth has led to environmental deterioration, especially excessive utilization and development of water resources, which has led to shrinkage of lakes, a reduction in floodplains, and loss of wetlands and other factors that are crucial to goose habitat [14,17,18,35]. The Chinese government has taken many measures to protect and restore the environment in recent years, such as returning farmland to wetlands and the Yangtze River protection strategy, and although the environment in some areas has been improved, the geese in the MLYR still face enormous challenges.

## 5. Conclusions

Based on SSDM and GPS tracking data, in this study, we analyzed the changes in the suitable goose habitat areas in the MLYR from 2000 to 2019. The results showed that the suitable goose habitat areas in this region had experienced varying degrees of loss, and that the suitable goose habitat area was significantly reduced. The LWFG had the greatest suitable habitat area loss (over 50%), while the GG had the least suitable habitat area loss (6.24%). GWFG, BG, and SG suitable habitat areas were reduced by 34.00%, 30.81% and 13.71%, respectively. The widespread and rapid changes in land use were one of the main reasons, especially the changes in floodplains, lakes, and wetlands. These analyses show that land use is an important factor in studying the spatial and temporal changes of suitable goose habitat areas on a large scale, which is of great significance to the protection and management of goose habitats.

**Author Contributions:** Conceptualization, K.H., J.L., Y.J. and G.L.; methodology, K.H. and G.L.; validation, K.H., Q.Z. and G.L.; investigation, K.H., J.L., Y.J., E.W., G.S. and Q.Z.; writing—original draft preparation, K.H. and G.L.; writing—review and editing, K.H. and G.L.; supervision, G.L. and C.L.; project administration, G.L. All authors have read and agreed to the published version of the manuscript.

**Funding:** This work was supported by National Natural Science Foundation of China (no. 31971400) and the National Key Research and Development Program of China (2017YFC0405303).

**Data Availability Statement:** Not available.

**Acknowledgments:** We would like to thank researcher Qiang Dai at the Chengdu Institute of Biology Chinese Academy of Sciences for assistance with models and data analysis.

**Conflicts of Interest:** The authors declare no conflict of interest.

## Appendix A

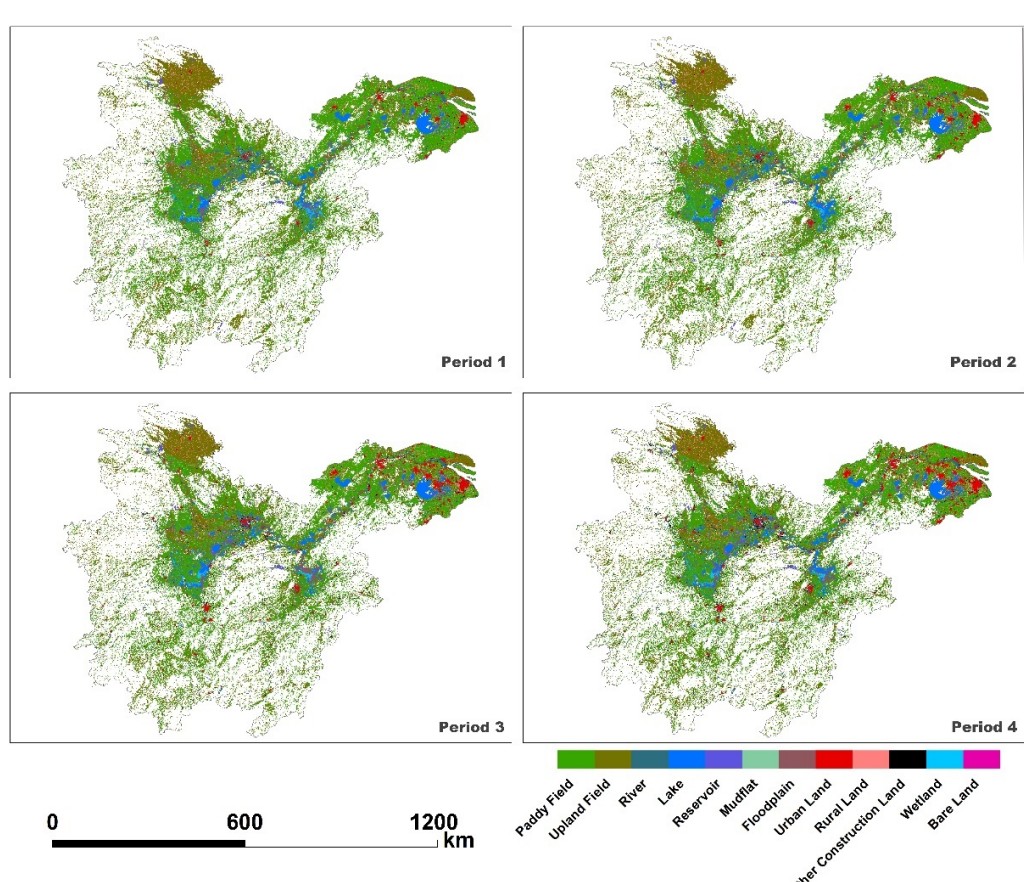

**Figure A1.** Land use in four periods. Period 1 is 2000; Period 2 is 2005; Period 3 is 2010; Period 4 is 2015.

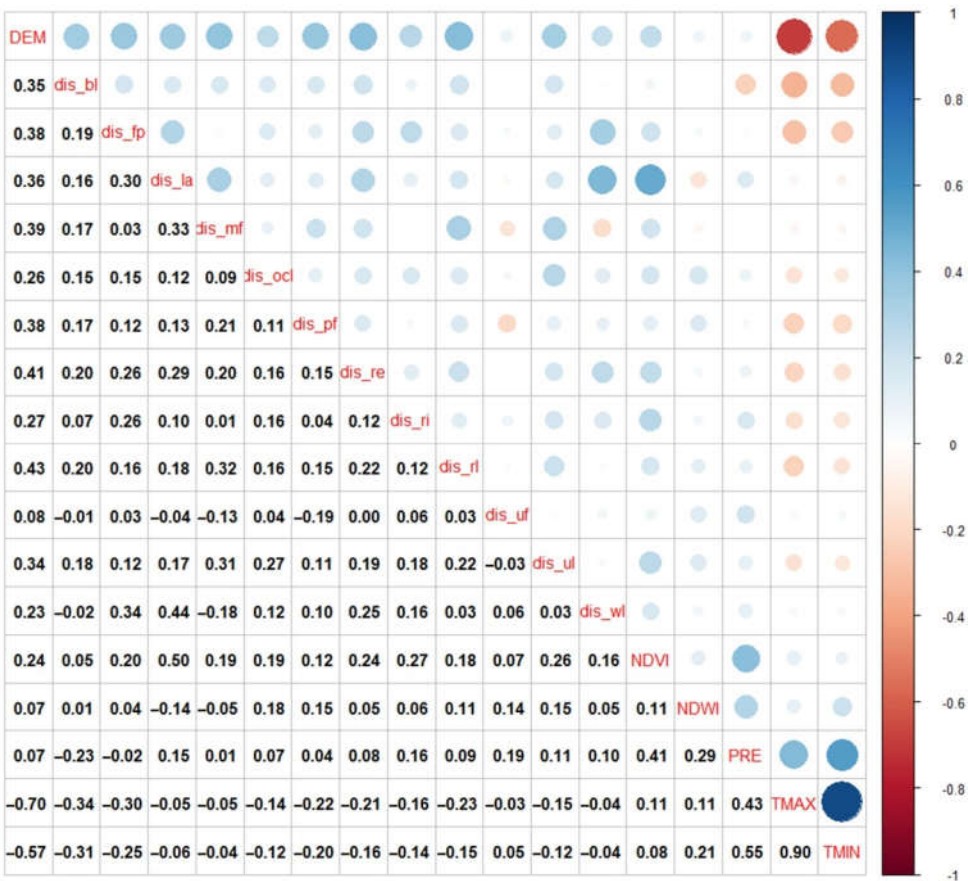

**Figure A2.** Correlation of 18 variables. (BL, bare land; DEM, altitude; FP, floodplains; LA, lake; MF, mudflat; NDVI, normalized difference vegetation index; NDWI, normalized difference water index; OCL, other construction land; PF, paddy field; PRE: monthly precipitation; RE, reservoir; RI, river; RL, rural land; TMAX, monthly mean maximum temperature; TMIN, monthly mean minimum temperature; UF, upland field; UL, urban land; WL, wetland). Red points represent negative correlations and blue points represent positive correlations.

## Appendix B

**Table A1.** Variable importance of 18 environmental variables of the GAM algorithm.

| Variables | LWFG | GWFG | BG | SG | GG |
|---|---|---|---|---|---|
| Dis_pf | 0.012 ± 0.009 | 0.020 ± 0.010 | 0.011 ± 0.012 | 0.017 ± 0.007 | 0.008 ± 0.009 |
| Dis_uf | 0.015 ± 0.010 | 0.014 ± 0.016 | 0.019 ± 0.008 | 0.026 ± 0.011 | 0.018 ± 0.004 |
| Dis_ri | 0.032 ± 0.013 | 0.029 ± 0.006 | 0.020 ± 0.010 | 0.012 ± 0.012 | 0.022 ± 0.013 |
| Dis_la | 0.065 ± 0.021 | 0.156 ± 0.051 | 0.158 ± 0.028 | 0.167 ± 0.051 | 0.277 ± 0.097 |
| Dis_re | 0.015 ± 0.011 | 0.036 ± 0.016 | 0.014 ± 0.013 | 0.037 ± 0.017 | 0.032 ± 0.031 |
| Dis_mf | 0.086 ± 0.026 | 0.109 ± 0.025 | 0.066 ± 0.021 | 0.062 ± 0.023 | 0.046 ± 0.022 |
| Dis_fp | 0.125 ± 0.036 | 0.116 ± 0.044 | 0.137 ± 0.031 | 0.144 ± 0.046 | 0.131 ± 0.060 |
| Dis_ul | 0.014 ± 0.009 | 0.012 ± 0.012 | 0.025 ± 0.013 | 0.027 ± 0.014 | 0.013 ± 0.011 |
| Dis_rl | 0.016 ± 0.007 | 0.026 ± 0.012 | 0.021 ± 0.007 | 0.025 ± 0.013 | 0.032 ± 0.029 |
| Dis_ocl | 0.013 ± 0.010 | 0.022 ± 0.013 | 0.016 ± 0.013 | 0.017 ± 0.008 | 0.010 ± 0.011 |
| Dis_wl | 0.259 ± 0.045 | 0.188 ± 0.053 | 0.088 ± 0.031 | 0.217 ± 0.079 | 0.175 ± 0.065 |
| Dis_bl | 0.069 ± 0.018 | 0.078 ± 0.015 | 0.044 ± 0.015 | 0.044 ± 0.025 | 0.024 ± 0.015 |
| DEM | 0.019 ± 0.026 | 0.036 ± 0.047 | 0.041 ± 0.037 | 0.036 ± 0.050 | 0.021 ± 0.026 |
| NDVI | 0.022 ± 0.007 | 0.040 ± 0.014 | 0.060 ± 0.010 | 0.059 ± 0.028 | 0.112 ± 0.035 |
| NDWI | 0.020 ± 0.008 | 0.021 ± 0.012 | 0.041 ± 0.019 | 0.024 ± 0.017 | 0.027 ± 0.025 |
| PRE | 0.014 ± 0.011 | 0.019 ± 0.010 | 0.038 ± 0.015 | 0.010 ± 0.008 | 0.010 ± 0.010 |
| TMAX | 0.082 ± 0.032 | 0.042 ± 0.021 | 0.094 ± 0.036 | 0.037 ± 0.021 | 0.023 ± 0.017 |
| TMIN | 0.123 ± 0.031 | 0.035 ± 0.028 | 0.108 ± 0.041 | 0.040 ± 0.017 | 0.019 ± 0.011 |

**Table A2.** The average of the three variables in the four periods.

| Period | Distance to Wetland | Distance to Lake | Distance to Floodplain |
|---|---|---|---|
| Period 1 | 128,104.553 ± 125.594 | 88,059.992 ± 101.239 | 18,951.819 ± 20.392 |
| Period 2 | 137,343.562 ± 128.758 | 83,960.312 ± 99.655 | 17,731.429 ± 18.043 |
| Period 3 | 156,412.35 ± 139.942 | 69,673.015 ± 74.446 | 17,362.222 ± 16.905 |
| Period 4 | 156,595.611 ± 140.027 | 61,968.91 ± 63.665 | 17,159.595 ± 16.493 |

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
