# Peer review of "Temporal Dynamics of the Goose Habitat in the Middle and Lower Reaches of the Yangtze River"

_remotesensing, doi:10.3390/rs14081883_

Round 1

Reviewer 1 Report

It's unclear if goose counts are declining or staying the same.

In lines 157 to 159, it is indicated that the resolution of the DEM was converted to 1 km squared. This supposes a decrease in resolution, a bad resolution for a DEM.

In lines 138 to 151, what is the resolution of Land Use?

On lines 223 to 237, the information in Figure 5 and Table 3 do not match. They are poorly expressed.

In lines 308 to 311, it indicates that the LWFG populations decrease to 14,000-19,000 individuals in 2016, according to the bibliographic citation of Jia et al. [58]. But that's not true. Jia's work was published in 2016, but the bird census is up to 2011. That is, it only represents around 50% of the study period of this revised work. Authors should search for other references and add them.

Reviewer 2 Report

Good for the publication in the present form

Author Response

The reviewer did not ask questions

Reviewer 3 Report

The article is interesting and its content is concise and clear.

Comments, mostly concerning a broader explanation of some elements of the methodology and some editorial flaws, have been placed next to the text of the article in the attached PDF file.

  1. Since land use turned out to be the main variables in modeling the habitats of five migratory species of geese, I believe that the article should include a map of the coverage / use of the studied area from 2000, as well as maps showing its changes in subsequent periods. The discussion of the results should include an excerpt about which changes, in the context of the obtained results, between the LULC categories are the most unfavorable for geese.
  2. Some conclusions resulting from the content of Figure 2, concerning the usefulness (effectiveness) of individual algorithms used for modeling, would also be valuable.

Author Response

This manuscript is a resubmission of an earlier submission. The following is a list of the peer review reports and author responses from that submission.